# Emerging Developments on Nanocellulose as Liquid Crystals: A Biomimetic Approach

**DOI:** 10.3390/polym14081546

**Published:** 2022-04-11

**Authors:** Theivasanthi Thiruganasambanthan, Rushdan Ahmad Ilyas, Mohd Nor Faiz Norrrahim, Thiagamani Senthil Muthu Kumar, Suchart Siengchin, Muhammad Syukri Mohamad Misenan, Mohammed Abdillah Ahmad Farid, Norizan Mohd Nurazzi, Muhammad Rizal Muhammad Asyraf, Sharifah Zarina Syed Zakaria, Muhammad Rizal Razman

**Affiliations:** 1International Research Center, Kalasalingam Academy of Research and Education, Anand Nagar, Krishnan Koil 626126, India; t.theivasanthi@klu.ac.in; 2School of Chemical and Energy Engineering, Faculty of Engineering, Universiti Teknologi Malaysia, Johor Bahru 81310, Malaysia; 3Centre for Advanced Composite Materials (CACM), Universiti Teknologi Malaysia, Johor Bahru 81310, Malaysia; 4Research Centre for Chemical Defence, Universiti Pertahanan Nasional Malaysia, Kem Perdana Sungai Besi, Kuala Lumpur 57000, Malaysia; 5Department of Mechanical Engineering, Kalasalingam Academy of Research and Education, Anand Nagar, Krishnan Koil 626126, India; tsmkumar@klu.ac.in; 6Department of Materials and Production Engineering, The Sirindhorn International Thai-German Graduate School of Engineering (TGGS), King Mongkut’s University of Technology North Bangkok, 1518 Wongsawang Road, Bangsue, Bangkok 10800, Thailand; suchart.s.pe@tggs-bangkok.org; 7Institute of Plant and Wood Chemistry, Technische Universität Dresden, Pienner Str. 19, 01737 Tharandt, Germany; 8Department of Chemistry, College of Arts and Science, Yildiz Technical University, Davutpasa Campus, Istanbul 34220, Turkey; syukrimisenan@gmail.com; 9Department of Bioprocess Technology, Faculty of Biotechnology and Biomolecular Sciences, Universiti Putra Malaysia, Serdang 43400, Malaysia; abdillah.upm@gmail.com; 10Center for Defence Foundation Studies, Universiti Pertahanan Nasional Malaysia, Kem Perdana Sungai Besi, Kuala Lumpur 57000, Malaysia; mohd.nurazzi@gmail.com; 11Institute of Energy Infrastructure, Universiti Tenaga Nasional, Jalan IKRAM-UNITEN, Kajang 43000, Malaysia; asyrafriz96@gmail.com; 12Research Centre for Environment, Economic and Social Sustainability (KASES), Institute for Environment and Development (LESTARI), Universiti Kebangsaan Malaysia (UKM), Bangi 43600, Malaysia; 13Research Centre for Sustainability Science and Governance (SGK), Institute for Environment and Development (LESTARI), Universiti Kebangsaan Malaysia (UKM), Bangi 43600, Malaysia; mrizal@ukm.edu.my

**Keywords:** nanocellulose, cellulose nanocrystals, liquid crystals, biomimetic

## Abstract

Biomimetics is the field of obtaining ideas from nature that can be applied in science, engineering, and medicine. The usefulness of cellulose nanocrystals (CNC) and their excellent characteristics in biomimetic applications are exciting and promising areas of present and future research. CNCs are bio-based nanostructured material that can be isolated from several natural biomasses. The CNCs are one-dimensional with a high aspect ratio. They possess high crystalline order and high chirality when they are allowed to assemble in concentrated dispersions. Recent studies have demonstrated that CNCs possess remarkable optical and chemical properties that can be used to fabricate liquid crystals. Research is present in the early stage to develop CNC-based solvent-free liquid crystals that behave like both crystalline solids and liquids and exhibit the phenomenon of birefringence in anisotropic media. All these characteristics are beneficial for several biomimetic applications. Moreover, the films of CNC show the property of iridescent colors, making it suitable for photonic applications in various devices, such as electro-optical devices and flat panel displays.

## 1. Introduction

Liquid crystals denote an intermediate condition between crystalline solid and amorphous liquid, making it a mesophase. Like a liquid, they have some degree of isotropy. Simultaneously, a crystal’s order can be preserved in at least one direction, resulting in anisometric molecules with no positional order and just orientational order [1]. The characteristic of rod-like liquid arrangement during the crystalline phase with liquid and crystalline solid is depicted in Figure 1. The liquid crystalline phase structure can be found in a variety of living organisms, including plants, animals, and microorganisms. For instance, cellular membranes are made up of a lyotropic liquid crystalline phase that forms when phospholipids dissolve in water. The curvature of the membrane is essential for its function, particularly in cell signaling and trafficking, as well as the mycobacterial lipids release [2].

Inspired by nature, the development of novel biomimetic functional systems that mimic both structural and adaptive properties of animals and living beings have received a lot of attention [3,4,5,6,7]. The bionic materials, biomimetic, or biomimicry are intended to mimic or replace the spatial and natural temporal presentations of physiological and biochemical components to restore or enhance biological functioning. The modeling of one or more biological systems’ functional principles is all that is required for biomimetic material design. One method for developing biomimetic concepts is to list the desired system’s fundamental activities first and then find a biological system that may serve as a model and execute comparable functions [8]. In regenerative medicine and regulated or target-specific drug delivery systems, rationally developed materials that investigate specific, customizable, biodegradable, and reversible interactions have benefits over non-modified standard materials. These materials are referred to as biomimetic when they are designed for detection and responding to physiological inputs or to mimic structural and functional elements of biological signals [9].

Nanomaterials derived from renewable resources are generating a lot of interest, both in fundamental science and building new structural and functional macroscopic materials [10,11,12,13,14,15,16,17,18,19,20,21,22]. The exceptional mechanical properties and/or photonic crystal features of biological composites, e.g., nacre, bone, beetle scales, wood, and butterfly wings, are also essential inspirations for developing novel multifunctional materials using nature-based nanomaterials. Nonetheless, to fully exploit the intrinsic features of nanosized starting materials, strong and adaptable synthetic and processing strategies to regulate assembly over many length scales must be developed [23].

For millennia, humans have been using cellulose, among nature’s most versatile and broadly distributed biopolymers, as a building material, source of energy, clothing component, and a means of storing and transmitting culture and knowledge [24,25]. Presently, cellulose products are used in a wide array of uses, and the pulp and paper industry contributes significantly to the economic output of many countries [26,27,28,29,30,31,32,33,34]. Cellulose is insoluble in water and possesses a hydrophilic surface. Cellulose contains both amorphous and crystalline regions [35]. Oxidation of cellulose may result in carboxylic acids from some hydroxyl groups conversion [36]. Nanocellulose is spindle-shaped with Young’s modulus of more than 140 GPa. Cellulose extraction can be achieved in various sizes, depending on the desired applications [33,37,38].

In industrial practices, nanocellulose is the most frequently used cellulose size. Nanocellulose is categorized into three classes: (1) nanofibrillated cellulose (NFC) or nanofibrils or macrofibrillated cellulose or microfibrils, (2) cellulose nanocrystals (CNC) or crystallites or rod-like cellulose microcrystals or whiskers, and (3) bacterial nanocellulose (BNC) or biocellulose or microbial cellulose [39,40,41,42,43,44,45]. CNC is prepared via cellulose acid hydrolysis containing natural materials like cotton and wood [46,47,48]. According to Kusmono et al. [49], sulfuric acid has been recognized as the acid most commonly used in hydrolysis for preparing CNC due to the process is simple and results in nanoparticles (100–1000 nm) with highly crystalline and stiff. During the acid hydrolysis process, the sulfur groups attach to the CNC surface and become negatively charged. Because of the electrostatic repulsion between the CNC particles, a good suspension is obtained. This can be analyzed by optical polarization microscopy for its liquid crystalline nature, which is suitable for more photonic applications [50].

NFC and CNC are distinguished by the distribution of wider fiber sizes in NFC and drastically shorter or narrower in CNC [51], as depicted in Figure 2. Both NFC and BNC have high crystallinity and narrow size distribution, except for BNC that is derived from bacteria. Alain Dufresne [52] and Chirayil et al. [53] reported the fame of CNC and NFC is not only due to their biodegradability, natural abundance, unique structures, superior mechanical properties, low density, high surface area and aspect ratio, and biocompatibility but also for their possibility to modify their surfaces to enhance their nano-reinforcement compatibility with other polymers due to the presence of abundant hydroxyl groups. Nanocellulose-based materials, also known as a new ageless bionanomaterial, are non-toxic, recyclable, sustainable, and carbon-neutral [52]. Nanocellulose has the properties of nanoscale dimension, low density, chirality, and thermomechanical performance. Due to these attributes, nanocellulose has gained tremendous attention from scientists. In consequence, the number of patents and publications on nanocellulose over 20 years has increased significantly from 764 in 2000 to 18,418 in 2020. In addition, this increment of more than 2300% over 20 years indicates that nanocellulose has become the advanced emerging material in the 21st century.

Assembling nanocellulose into multiphase structures leads to many advanced applications. Nanocellulose fibers contain both crystalline and amorphous phases. The OH groups’ abundance on nanocellulose promotes the formation of hydrogen bonds, resulting in highly ordered cellulose chains. They form networks when dispersed in water. Biopolymers are abundant, renewable, and sustainable [56]. By utilizing nanocellulose’s colloid and interface properties, it can be formed into emulsions, solid films, foams, and aerogels. Nanocellulose self-assembly into a chiral nematic liquid crystalline phase can be applied in functional films. In concentrated dispersions, nanocellulose constructs cholesteric ordering and arranges as a chiral nematic liquid crystal. The nanocellulose films reflect the polarized light. The film color can be varied by varying the film thickness. A multilayer system can be constructed by alternate deposition of nanocellulose and a cationic polymer. Ultrasound treatment can change the chiral nematic pitch. Sonication can control the iridescent color. Externally applied electric and magnetic fields can affect the orientation. Nanopapers or nanocellulose films have good gas barrier properties since the porosity is low.

Therefore, this review aims to discuss the current research on developing nanocellulose-based solvent-free liquid crystals and their utilization in photonic applications. In addition to that, the isolation and production of CNC-based liquid crystals together with their biomimetic application were also tackled. Lastly, some future recommendations related to this study are also suggested.

## 2. Isolation of CNC from Several Biological Sources

In nature, cellulose exists in the form of the smallest size of an elementary fibril, regardless of the source [57,58]. There are two key steps in extracting nanoparticles from a raw cellulose sample, including (i) the source material purification and homogenization to allow homogeneous reaction conditions and (ii) the purified cellulose material separation into its components of microfibrillar and/or nanocrystalline [59].

Depending on the cellulose source, the first step is adjusted. The matrix elements removal from lignin and hemicellulose is the most critical phase for wood and plants. Luzi et al. developed a CNC extraction method from the North African grass *Ampelodesmos mauritanicus*, or Diss. The stems of the Diss were subjected to two distinct pretreatments before being extracted for nanocellulose (chemical or enzymatic). In the first pretreatment stage or the chemical pretreatment, sodium bisulfate (NaHSO_4_) was employed to remove holocellulose (-cellulose + hemicellulose). The following pretreatment included the extraction of the -cellulose component and elimination of hemicellulose using sodium hydroxide (NaOH). Xylanase (Feedlyve AXC) and polygalacturonase (Peclyve EXG) enzymes were utilized in the enzymatic pretreatment. Both pretreatments aided in the reduction of fiber diameter, according to morphological analysis. New crystalline domains were created as a result of the enzymatic breakdown of cellulose, according to X-ray diffraction spectra. When compared to enzyme-treated CNC, chemically-treated CNC had lower values of mean diameter and length, as shown in Figure 3.

In the second process step, acid hydrolysis, mechanical treatment, and enzymatic hydrolysis are the approaches that can be used to separate the pretreated fibers into microfibrillar or nanocrystalline components [61]. The most frequently used CNC isolation from cellulose fibers technique is acid hydrolysis, a method that dates back to Ranby’s foundational studies [62], with a number of more recent modifications. The characteristics of CNC from several sources separated via sulfuric acid hydrolysis are summarized in Table 1.

The use of enzymes in the manufacture of CNC is complicated, and they can play a variety of functions in the entire process, from eliminating pectin and hemicellulose to the most prevalent use of cellulases to form CNC. Despite the widespread belief that CNC is produced primarily through acid hydrolysis, numerous studies have shown the CNC materials production mechanisms are more varied; for example, multiple works mentioned the use of cellulases in conjunction with acid hydrolysis to produce CNC. A summary of prior research has noted the generation of CNC via enzymatic hydrolysis methods and tabulated in Table 2.

Lerma et al. [73] produced performance crystalline CNC using ancestral endoglucanase. Enzymatic hydrolysis was conducted in water at 50 °C in agitation with several endoglucanases at various times. The hierarchical structure of cellulose fiber allows for nanoscale particle destruction. They demonstrated how endoglucanase with a carbohydrate-binding module (CBM) could bind to cellulose fibers, embedding the catalytic domain in the fiber surface and increasing activity. Depending on the hydrolysis duration and enzyme used, enzyme degradation can yield nanocellulose of various sizes. The CNC is a small crystalline particle with diameters ranging from 3 to 40 nanometers and lengths ranging from 100 to 500 nanometers. Figure 4 shows the illustration of cellulose enzymatic hydrolysis.

## 3. Overview of CNC-Based Liquid Crystals

Revol et al. were the first to uncover the structural feature of CNC-based liquid crystals. CNC’s structural property allows it to diffuse freely in water and self-organize spontaneously into a cholesteric liquid crystals phase at a low mass fraction [74]. Lower concentrations of CNC in suspension water are described to assemble isotropically (Figure 5). The suspension changed from dilute to semi dilute and concentrated isotropic assembly as the concentration rose. The CNC began to organize as tactoids above a certain concentration, which are basic phases order of liquid crystalline that forms later. During the biphasic phase, each of the liquid crystalline and isotropic order domains was in balance with the other. A higher concentration results in excellent liquid crystalline order, and the shift from the liquid crystalline to the gel state is a complex interaction of interparticle attraction and concentration. Electrolytes can compress the electrical double layers, which lessens the repulsion of charged particles and, according to reports, causes a shift between liquid and solid phases [75].

On the other hand, the anisotropic structure creation within a soft solid suspension is feasible when the isotropic-nematic and liquid-solid transitions coincide. The liquid crystal re-entrance, as well as the presence of an anisotropic soft solid, resulted from research into these two competing transitions (liquid crystals hydro glass, LCH). LCH is a biphasic material composed of an appealing glass matrix and a coexisting liquid crystals phase that exhibits similar viscoelastic behavior to hydrogels while allowing colloidal rods to be reversibly oriented in the liquid crystalline phase via shear forces, e.g., their structural ordering is programmable [76]. This is depicted in Figure 6 by the isotropic/anisotropic behavior of liquid/solid phases. The existence of a reentrant liquid crystals phase at a high ionic strength satisfies the conditions for the development of LCH: spinodal breakdown (and subsequent gelation) coinciding with an anisotropic liquid crystals phase. The schemes A–D in Figure 6 illustrate the interaction between liquid crystallinity and gelation in CNC suspensions using the phase transition of a 7 wt% CNC suspension containing NaCl as an example. When salt was gradually added to a liquid crystals suspension (A), the liquid crystals’ phase proportion decreased at first (B) and then increased (C). The suspension was subsequently “quenched” into the unstable thermodynamic zone (D), resulting in spinodal breakdown and segregation into the DL and DS phases, yielding a biphasic LCH. The matrix of LCHs was formed by DS, a colloid-rich phase with an appealing glass shape that gives solid-like rheological behavior. DL is a liquid crystals phase that gives anisotropic properties to LCHs.

Additionally, chemically produced aqueous suspensions of CNC, avoiding electrostatic stability and encouraging steric interaction, resulted in mesophase (Figure 7) with the usual fingerprint texture [80].

A cholesteric phase is a type of nematic phase in which the nematic phase’s nanoparticle structure is chiral. One intriguing practice of the CNC solution exhibiting the cholesteric phase is the ability to dry the mixture while sustaining the chiral structure in order to produce photonic bandgap films. The chiral structure has a lot of optical activity along its helix axis. In dried CNC films, this helical structure demonstrates significant birefringence. The reflected light wavelength causing color transition is determined by the value of P, also confirmed on a CNC-based liquid crystals film. The Bragg (Vries 1951) equation describes the wavelength of reflected light, given as Equation (1):λ = nPsinθ(1)
where λ denotes the reflected wavelength, P represents the helical pitch, θ is the incident light angle (°), sin θ = 1 when the incident light is perpendicular (90°) to the crystal plane, and n is the average refractive index of the material.

CNC aligns parallel to the director during this cholesteric phase, and its position spins through the liquid crystals along an axis perpendicular to the director, creating the helix (Figure 8a,b). De Vries devised a theory in 1951 to explain the relationship between the optical rotation and helix pitch. CNC films selectively reflect left-handed circularly polarized light due to their chiral nematic structure, resulting in a positive circular dichroism (CD) signal for transmitted light (Figure 8c,d).

## 4. Formation Process of CNC-Based Liquid Crystals

### 4.1. Self-Assembly of CNC

According to Onsager’s theory, the nematic ordering of rod-like CNC is determined by the balance of translational and orientational entropy. Schütz et al. reinterpreted “excluding volume” to better comprehend those entropic contributions that produced nematic ordering: the space is inaccessible for a rod due to the presence of other rods. This reduced orientational entropy and limited freedom of motion, but the total of excluded volume was limited, and translation freedom was improved by aligning themselves in a single direction, boosting translational entropy, and lowering free energy. Additionally, CNC aqueous solution can also demonstrate good stability against agglomeration due to electrostatic repulsion, which is favorable to the development of the phase of the liquid crystal.

#### 4.1.1. Vacuum-Assisted Self-Assembly (VASA)

In 2020, Wang et al. discovered the formation of the CNC chiral nematic phase via the vacuum-assisted self-assembly (VASA) method. Initially, hydrostatic forces deposited the distributed rod-like CNC on the filter membrane randomly and fast (30 s), generating a film-like gel that reduced the flow rate of water. As a result of the slower filtering velocity, the CNC had more time to self-assemble into liquid crystals structures with helical axes orientated along the hydrostatic force direction under the greater vacuum force [82]. As filtration proceeded, the number of tactoids increased until they coalesced into a long-range periodic chiral nematic structure [83]. Concurrently, the entrapment influence on the initial layer of constructed gel, combined with hydrostatic and electrostatic forces, compacted the final structure parallel to the interface. The procedure was then repeated until all of the surplus solvents were eliminated [84]. The vacuum-assisted CNC self-assembly method is depicted schematically in Figure 9.

In addition, a suspension of 8.5% of cellulose nanocrystal can be used to prepare films with chiral nematic liquid crystals’ optical properties. These films circularly reflected polarized light with a narrow wavelength that varies with the angle of viewing. The color of these films’ reflections can be adjusted by adding salt to the suspension. The CNC self-assembly into chiral nematic liquid crystalline structures did not only occur in aqueous but also in the dry state. They constructed a chiral nematic phase at high concentrations, a characteristic of liquid crystalline polymers, giving rise to attractive optical properties [85]. In the dry state, the crystalline phase of chiral nematic liquid can be preserved. In addition, the color formation dynamics can also be varied by altering the film formation environment [86].

#### 4.1.2. Evaporation Induced Self-Assembly (EISA)

Evaporation-induced self-assembly is a method of synthesis where a drop of solvent is applied to the elements to be assembled and left to dry naturally in the air. Individual components spontaneously associate into an organized pattern or structure as a result of solvent evaporation. The maximum reflection wavelength is related to the pitch P using Braggs’ formula, λ = navPsinθ, where nav denotes the average refractive index of the crystalline material (1.56 for CNC), and θ represents the incidence angle of circularly polarized light on the film surface [87]. The quality of the film depends on CNC orbital shear, concentration, drying conditions, and surface anchoring. CNC films of large planar domains were fabricated by slowly drying CNC dispersions placed between two glass surfaces in a water vapor-saturated environment on an orbital mixer. These domains exhibited the double peaks reflectance spectra biomimetic feature like in Lomaptera beetles [88].

The liquid crystalline phase emergence was observed by Jativa et al. in the constrained space of an isolated and shrinking CNC-containing droplet soaked in a binary toluene–ethanol mixture (Figure 10). The shrinking droplet’s contact angle with the hydrophobized substrate was so high (155°) that it resulted in the preservation of the droplet’s spherical shape throughout the drying process [89].

#### 4.1.3. Functionalization of CNC for Self-Assembly

Nanocellulose in its natural form has limitations in several applications, including as a biomimetic. Modification via surface functionalization is a critical step in enhancing biocompatibility functions in a variety of applications. This can be accomplished through a variety of surface functionalization strategies, the majority of which incorporate the chemistry of hydroxyl function. Additionally, as illustrated in Figure 11, a search on lens.org using the keyword “functionalization of nanocellulose” revealed an increase in the number of manuscripts putting the focus on nanocellulose functionalization in recent years. As a result, this demonstrates that investigation on nanocellulose functionalization has maintained a high level of interest among scientists over the last decade.

For the CNC functionalization, usually, 12% acrylamide was grafted on CNC, which self-assembled into chiral nematic suspension at 3% content [90]. The liquid crystals properties analysis was performed using polarizing optical microscopy. The acrylamide grafted nanocellulose was self-assembled to a lyotropic state. The formation of liquid crystals’ structure was because of the attractive Van der Waals forces and repulsive electrostatic forces [90]. An electrolyte like NaCl was added to CNC suspension before evaporation took place to produce colored films. CNC films can also be produced using spin coating on mica wafers. The electrolyte addition like NaCl to chiral nematic CNC suspensions reduces the pitch values to prepare iridescent films of the desired wavelength.

The CNC films’ colors can be changed by adjusting the chiral nematic pitch P of cellulose nanocrystal. The addition of glucose increased the pitch and shifted the color to the spectrum’s red end in the final film [91]. Grafting the CNC with a charged organosilane and functionalizing using a counter-ion polyoxyethylene ether resulted in the formation of liquid crystals nature. This suspension possessed a bright birefringence between the crossed polarizers at room temperature [92]. Figure 12 shows 6% CNC suspension as viewed from polarized light microscopy [93].

### 4.2. Sacrificial Templates

Sacrificial templating is the most extensively utilized technique for generating nanocellulose-based porous materials because of the variety of fabrication processes that fall under this category, as well as the range of pore sizes/morphologies that may be achieved. This method entails creating a template within a nanocellulose solution or gel that may then be varied to generate a variety of pore morphologies and then eliminated using different methods. In most cases, a solvent, e.g., water, is employed in sacrificial templating, although care must be given to avoid pore collapse when the solvent is removed. As a result, the supercritical or freeze-drying method is frequently utilized to assure pore stability in final porous materials. In reality, the used drying procedure is significantly affecting the shrinkage, porosity, density, and pore size. Specific surface area, interconnectivity, and size distribution of the material: smaller pores are frequently the result of supercritical drying (usually micro and mesopores) versus freeze-drying, resulting in macroporous analogs [94].

These chiral nematic structures are embedded into silica, organo-silica, hydrogels, thermoset phenol-formaldehyde, and amino formaldehyde resins. Selective removal of nanocellulose can be used as a sacrificial template to produce materials like mesoporous silica. These types of mesoporous materials are useful in stereospecific catalysis, chiral separation, photonic materials, and chiral recognition (sensing). By minor changes in the synthetic environments, the peak reflected wavelength of the free-standing films could be differed across the whole visible and into the near-infrared spectra [95].

Introducing porosity into photonic crystals can tune their optical properties. The lyotropic liquid crystals self-assembly has been employed to introduce mesoporosity into photonic crystals, where these materials are used in the refractive index-based photonic sensors. CNC can be used as a template to create mesoporous structures. The CNC removal from CNC/silica composite was performed by calcination or by acid hydrolysis [96]. A composite of CNC and a urea-formaldehyde resin was formed, which underwent rapid and reversible color changes upon swelling can be used for pressure sensing application. At 700 °C, alkaline treatment of the composite films with 15% aqueous KOH removed the urea-formaldehyde, resulting in insoluble and luminescent cellulose films (mesoporous photonic cellulose films) [97]. The preparation of chiral nematic mesoporous carbon was performed via polymerization of polyacetylene within thermotropic chiral nematic liquid crystals, doping with iodine, and pyrolysis, displaying attractive electromagnetic characteristics [98].

## 5. Formation of Flexible CNC Films

To expand the applications of CNC, CNC is dispersed in organic solvents. These nanoparticles exhibit lyotropic liquid crystalline behavior in suspension, which refers to phase changes from an isotropic liquid to an ordered liquid crystal as the concentration changes. Indeed, a chiral nematic phase form exists above a certain concentration; meanwhile, within certain conditions, the suspension can be slowly evaporated, forming semi-translucent films that retain the order of the chiral nematic liquid crystals formed in the suspension. These films demonstrated iridescence when reflecting polarized light in a narrow wavelength range as determined by the films’ refractive index and chiral nematic pitch. These aforementioned CNC’s optical behaviors are potentially generating wider CNC films applications. Ultrasound treatment was found to increase the chiral nematic pitch in suspension and red-shift the reflection wavelength of CNC films as the applied energy increased.

In the film formation, CNC is first treated with a base and freeze-dried. This broadens the application possibilities for novel photonic materials prepared via CNC templating. Tape casting, also known as knife coating and doctor blading, is a process where a thin layer of ceramic slurry is cast onto a flat surface, dried, and sintered, which can also be used for preparing CNC films. This method is very effective in aligning CNC. Variables like pH, drying conditions, type of substrate, and CNC concentration are tuned to achieve different orientations in CNC films. Thus, CNC films exhibit both ceramics-like as well as polymer-like properties, which are based on film orientation. The color of CNC films is affected by changing the helical pitch, and tweaking this structural color has been an ongoing topic of interest for several research groups in the previous decade.

The films formed with CNC are usually brittle. To increase the films’ flexibility, some synthetic polymers can be added. The addition of polyethylene glycol (PEG) increased the flexibility of CNC film with a doubling of the elongation, and the color of the film was preserved [99]. Self-organization resulted in Bragg’s reflection of visible light from the dried films. Iridescence and colored films are only possible if the pitch length of the solid films is equivalent to the visible light wavelength [88].

Glycerol can be used as both a hygroscopic agent and plasticizer to produce controllably iridescent, flexible, and multi stimuli-responsive CNC film. The CNC self-assembly in an aqueous solution was not affected by the additive but made it free-standing. These films can change color structurally in response to mechanical compression and environmental humidity, where the colors were due to the interference of light in intermittently layered or lattice structures. These films find applications in anti-counterfeiting technology, colorimetric sensors, as well as decorative coatings [100]. Zwitterionic surfactants with nanocellulose can form flexible and iridescent films. Zwitterionic surfactants can connect neighboring CNC. When the ratio of the Zwitterionic surfactant was increased, there was a shift towards a higher wavelength [101].

The CNC/waterborne polyurethane (WPU) composite papers showed optical responses to wet gas and water. Thereby, chiral nematic structured CNC-derived photonic papers with rapid stimulus-response, rewritable efficiency, and high flexibility are prepared. The chiral nematic structure retention is very sensitive to ionic strength and pH value [102]. The CNC and glycerol are added in different compositions of 100/0, 90/10, 80/20, 70/30, 60/40, and 50/50 to produce intelligent, responsive iridescent films. The glycerol addition enhanced the mechanical performance and crystallinity of the composite films [103]. The CNC/PEG (80/20) composite film was used as a humidity sensor, which showed green to transparent color change for relative humidity (RH) between 50 and 100% due to the reversible chiral nematic structure’s dehydration and swelling [103].

In contrast, self-assembly of the chiral nematic consists of pseudo layers, where the CNC was aligned along with a director, and each director’s orientation was slightly rotated about the helicoidal axis from one layer to the next. The pitch (P) is the vertical distance necessary to finish a 180° rotation of the director. These films reflected only left-hand circularly polarized light and exhibited distinct differently colored domains. Scanning electron microscopic (SEM) images confirmed that the pitch is well-defined within the domains of a single color [85]. The nanocellulose films’ optical properties were analyzed in the wavelength range of 200–1000 nm for the regular light transmittance with a UV-visible spectrometer. The densely packed cellulose nanofibers resulted in films that were optically transparent; hence, the interstices between the fibers were sufficiently small to prevent light scattering. The 600 nm transmittances of soft and hardwoods TEMPO-oxidized films were observed at approximately 90% and 78%, respectively [52].

## 6. CNC-Based Liquid Crystals in Biomimetic Applications

Nature-based nanomaterials like nacre, bone, wood, butterfly wings, and beetle scales inspire man to apply the biomimetic approach to develop new multifunctional materials [23]. Nature’s brightest and most colorful materials, such as peacock feathers, butterfly wings, and opals, derive their color exclusively from their internal structure. Self-assembly of CNC to a chiral nematic (cholesteric) liquid crystals phase with a helical arrangement forms films with a photonic bandgap. When immersed in water, the rod-like CNC self-assembled into nanostructured layers that reflected the light of a particular color selectively. The reflected color is influenced by the layers’ dimensions. Varying the humidity during film formation changes the color [99]. Bio-inspired photonics is the genesis of the development of novel and multifunctional photonic materials. CNC has been in the limelight to produce nanostructure crystals-based photonic materials that yield iridescence [104]. Additionally, manipulating the spiral orientation of CNC may result in an improvement in mechanical qualities [105].

### 6.1. Photonic Applications of CNC

There are several photonic structures in nature that produce bright colors and serve as inspiration for the development of a broad range of artificial photonic materials. Unlike colorant-based pigmentation, structural colors result from light interference in periodic layered or lattice structures seen in insects, fish, birds, fruits, and leaves. Photonic crystals, Bragg stacks, and chiral nematic liquid crystals have all been explored in the last several decades to generate artificial structural colors based on these fascinating phenomena [106]. Because of their iridescence properties, CNC-based optical materials have the potential to replace hazardous dyes in cosmetics, food, art, textiles, security indicators, and sensors, among other applications, in the near future.

Natural photonic nanostructures, e.g., CNC liquid crystals, have the potential to generate vibrant colors. Since the insertion of sulfate ester groups to the surface of this compound, it has been well dispersed in water, and the electrostatic repulsion that results from this allows the creation of chiral nematic liquid crystals to occur [107]. To fabricate solid films with changeable iridescent colors, this liquid crystal may be employed that uses the helical pitch to position it in the visible wavelength area when an electric field is applied (Figure 13) [108]. This model suggests that the helical pitch orientation is critical in determining the incoming light’s maximum reflected wavelength [109]. The CNC can easily arrange into photonic structures with the correct helical orientation, resulting in the development of advanced optical materials.

To produce highly ordered superlattices and helicoidal structure networks whose pitch is in the visible wavelength range, self-assembly of colloidal nanocrystals following solvent evaporation is critical (Figure 14a) for CNC iridescent film fabrication [110]. Previously published work has shown that the iridescence of CNC films may be tuned by altering the pitch of the CNC dispersion through sonication to control the helical orientation of the chiral nematic phase [111]. Before film casting, ultrasonic energy increases the chiral nematic phase suspension pitch and shifts the resulting iridescent film’s reflection band to long wavelengths (Figure 14b). In the sample, the energy input per gram of CNC (degree of sonication) affected the amplitude of reflection band red-shift (reflection wavelength), and these effects of sonication on treated CNC suspensions were certainly permanent and cumulative. Figure 14c shows the evolution of the film’s conductivity and peak reflection wavelength as a function of the applied ultrasonic energy for a CNC suspension. An electrolyte added to the solution reversed the sonication’s effects, causing a blue shift in the reflection wavelength that could be used to control the film’s reflectivity. Surface charge density and other CNC parameters like particle length do not seem to have an impact on this effect [111].

In Chen and colleagues’ research [109], the iridescent chroma of CNC films may be controlled by sonication duration, suspension volume, and degree of vacuum. Chen and colleagues discovered that films made from CNC suspensions with longer average particle lengths reflect more redly, whereas films made from shorter CNC particles reflect more bluely. Given that sonication has the capability of shortening the CNC particles, it increases the pitch of the chiral nematic phase, causing a red shift in the reflection wavelength of the resultant solid iridescent film. When the sonication time exceeds 10 h, a distinct iridescent color is shown in Figure 15a. Longer sonication increases the brightness of the resultant film, as well as its color. While samples treated for short periods of sonication (7 and 10 h) did not exhibit any noticeable reflection peak in the wavelength range of 300–800 nm of UV-Vis spectra, longer periods of sonication (i.e., 12 and 15 h) revealed reflection peaks, which were blue-shifting with increasing sonication time. A linear connection between the film thickness and the volume of the CNC suspension is seen in Figure 15b. This suggests that the volume of the CNC suspension may be carefully regulated to get precisely controlled thicknesses of iridescent films (as the film’s thickness may attribute to iridescence). With increasing vacuum pressure (from 0.07 to 0.04 MPa), the iridescent color varies, as seen in Figure 15c. This finding implies that the iridescent color is likewise vacuum dependent.

Using a glycerol-derived plasticizer and a hygroscopic ingredient for CNC, He et al. [106] developed flexible and multi-responsive chiral nematic films that were inspired by the color shift of longhorn beetles *Tmesisternus isabellae* from golden in the dry state to red in the wet state. The quantity of glycerol used changed the iridescent color of these films from blue to red. These CNC/glycerol films were photographed vertically, as shown in Figure 16a, where distinct iridescent colors can be observed as the concentration of glycerol increases. This has been linked to the fact that light of a certain wavelength is lost when it passes by these films. By using UV-Vis, it was found that the addition of glycerol results in a progressive red shift of the maximum extinction wavelength from 347 nm to 610 nm (Figure 16b). Furthermore, the inclusion of glycerol gives the CNC films a high degree of flexibility, allowing them to be bent and stretched with an elongation-at-break exceeding 2% (Figure 16c). Reversible color changes are seen in the films when they are subjected to a range of relative humidity levels (Figure 16d). In light of the aforementioned characteristics and functionalities, it is anticipated that CNC films will find applications in colorimetric sensors, anti-counterfeiting, and aesthetic coatings.

CNC’s macroscopic characteristics are greatly influenced by its microscopic characteristics (rheology, colloidal stability, etc.). Modifying and functionalizing CNC are likely to improve colloidal stability with the addition of a surfactant or surface functionalities, which leads to the development of new and better materials with advanced capabilities. The inclusion of hydroxypropyl cellulose (HPC) into a CNC film made by slow evaporation was investigated by Walters and colleagues [112]. Changes in HPC concentration or molecular weight modulated the color over the visible spectrum (Figure 17a), and composite films were more flexible than neat CNC film, with a tenfold increase in elasticity, but decreased in stiffness and tensile strength (Figure 17b–d).

An optically transparent clear paper with >90% transmittance and haze value <1% has also been discovered [113]. Transparent cellulose paper is a photonic material for flexible electronics, optoelectronics, and photonics with the potential to replace SiO_2_. Mesoporous materials are the materials through which photons can be manipulated. Mesopores can be filled with high refractive index materials to create photonic structures. When cellulose nanofibers are densely packed, this transmittance >90% and haze value <1% can be achieved. Nano welding of fibers in ionic liquids gives much denser packing. With graphene oxide/cellulose crystal composites, proximity sensors with a faster response time and increased sensitivity can be developed. The composite’s high optical transparency and sensing capability enable it to be used in touch and non-touch screens in portable entertainment units, mobile phones, human skin, robotics, and explosive environments while consuming little energy.

Thermo-responsive photonic films with CNC and poly diol citrate elastomer have also been discovered by Espinha et al. (2016). Although cellulose-derived photonic structures possessed intense color, they demonstrated brittle structure. Poly diol citrate provided flexibility and enhanced functionality by giving the shape memory effect. Other polymers like polycaprolactone and PEG can also be used for this purpose. The materials integrated mesoporous photonic cellulose’s chiral nematic structural properties with the magnetic characteristic of cobalt ferrite. The dielectric feature significantly improved as a result of the mesoporous films’ effective swelling ability, an interesting feature for the application of electromagnetic interference shielding.

CNC-based aerogels are tough and flexible mechanically, optically transparent, and highly porous. They are suitable for applications requiring a fire-resistant, direction-dependent mechanical or electrical insulator. Birefringent inks made from CNC are applied in optical authentication and security printing. When printed on dark paper, these inks produce letters that are darker than the background without polarizers and brighter than the background with crossed polarizers [114]. Multifunctional biomimetic materials and sensors can be constructed using CNC/graphene composites. The color of the film can be altered by hydration or dehydration, similar to how beetles *Tmesisternus isabellae*’s elytra respond to water. The cracking and coffee ring effect of the CNC film can be avoided by the process of vacuum-assisted self-assembly. CNC coatings are made with aqueous CNC solutions that have applications in several colorants, UV-curable water-based clear coating formulation, and specific additives in clear coatings for wood. CNC liquid crystals membrane can be obtained by evaporating the cellulose nanocrystal at 20–80 °C for 1–12 h. This membrane is applied in wide practices, such as decoration, anti-fake labels, color coatings, and temperature sensors or optical verification equipment.

### 6.2. Hydro-Responsive Materials

In plants, bending and unbending motions caused by humidity constitute motivation for the development of new biomimetic materials. Even in the tissues of dead plants, similar motions have been documented. The development of anisotropic cellulose-based micro/nanostructures are engineered to alter form as environmental conditions change, forming the basis of the responsiveness of these materials [115]. Humidity-responsive, bioinspired, CNC-based sandwich assembled composite photonic film with a left-handed chiral nematic structure was reported by Wu et al. (2016). The asymmetric composite film was fabricated by stacking hydrophilic CNC and poly(ethylene glycol) diacrylate (PEGDA) layers, with the polyamide-6 layers being oriented in a uniaxial direction. Asymmetric swelling of the hydrophilic sandwich components, namely CNC and PEGDA, caused the cellulose composite film to contract once subjected to humidity. These results might be explained by the fact that an increase in helical pitch near the moisture source causes an asymmetrical expansion of sandwiched structure nanocomposite film (Figure 18), hence allowing swelling on the stimulated side but no changes on the opposite side.

### 6.3. 4D Printing Materials

4D printing is a one-step process in which several properties of materials can be combined to allow the conversion of a 1D strand or 2D surface into a 3D shape, or even the transformation of a 3D shape into another 3D shape, using, for example, only water as the activator. 4D printing is becoming increasingly popular as a method of prototyping new materials. Using this printing process, manufacturers are able to produce and manufacture intelligent, reconfigurable materials that are better suited to specific uses [117].

3D printing of renewable building blocks, for example, CNC, provides an appealing method of producing environmentally friendly materials. Anisotropic CNC-based viscoelastic inks were developed by Siqueira and colleagues to provide direct ink writing patterning of 3D objects [118]. The printing efficiency has been shown with a monomer-based ink that is constituted of CNC dispersed in a solution including 2-hydroxyethyl methacrylate (HEMA) monomer, polyether urethane acrylate, and a photo-initiator, which is cured by ultraviolet light. Anisotropic components were aligned and stiffened along the printing direction in structures printed using CNC-based inks, demonstrating shear-induced alignment of the anisotropic elements (Figure 19). As a result of these characteristics, CNC-based inks are promising candidates for biomimetic 4D printing of programmable reinforced materials that adapt to environmental stimuli.

### 6.4. Bone Substitute Materials

It has been decades since researchers started experimenting with synthetic materials that match real bone composite in chemical and structural features [119]. Therefore, new varieties of 3D scaffolds have been developed in response to the intrinsic relationship between hydroxyapatite (HAP) and human bone to address the shortcomings of presently used bone replacements [120].

Huang et al. [121] created a novel nanocomposite scaffold by homogeneously depositing hydroxyapatite atop a cellulose nanocrystals matrix floating in simulated body fluid (SBF). The HAP content of the nanocomposite may be adjusted from 15% to 47% by varying the pH of the SBF. Through the use of the directional freezing (DF) method and lyophilization, the nanocomposites were freeze-casted into porous scaffolds. According to the results of compression tests on the HAP/CNC foams, compared to standard freezing procedures, this solidification approach significantly improved mechanical qualities because of the unique orientation and anisotropic porous structure. Ice crystals grew vertically in response to the DF temperature gradient, forcing them to grow vertically. The procedure created a channel structure that was directed and continuous in respect to the compression force (Figure 20). The scaffold with a high HAP concentration had demonstrated better mechanical and thermal capabilities, suggesting that it may have the potential for use in bone tissue engineering.

### 6.5. Nanocomposites

The relationship between mechanical properties of natural fibers and nanoscale structural orientation of CNC is of particular interest from the perspective of polymer nanocomposite processing because they provide an opportunity to control fiber mechanical properties by tailoring the self-assembly of nanoparticles within the polymer matrix. Research has shown that this structure–property relationship appears in CNC reinforced fibers. The mechanical characteristics of the fiber may be adjusted by regulating the CNC weight fraction loading and the fiber wet-spinning qualities. The mechanical properties of the fiber are closely related to the CNC spiral angle inside the fiber.

Using a biomimetic approach, Esteban and colleagues [105] developed polymer nanocomposites with controllable structure–property relationships. Adding CNC to alginate fiber increases the modulus at high loading CNC, as shown by the nanocomposite tensile characteristics for constant *J_A_*. These results imply that this phenomenon might be a result of the creation of a CNC percolation network across the fiber. Even at modest loads, adding CNC improved fiber toughness and elongation at break hence allowing for wet spinning of nanocomposite fibers to be carried out at a higher *J_A_*, doubling the fiber output rate. When the CNC concentration is high enough, the spiral angle seems to develop; however, higher levels of *J_A_* hinder the spiral structure’s emergence but encourage alignment along the fiber axis instead.

Another study by Fernandes et al. [122] introduced a low loading of CNC into liquid-crystalline HPC, resulting in a significant increase in mechanical features without compromising the composite liquid-crystalline order. There were two similar sets of bands detected in low concentration CNC/HPC films, which were studied by AFM (Figure 21a–c). It concluded that while CNC facilitates the reinforcement of the composite, the development and periodicity of the bands are primarily dictated by the chiral nematic-liquid-crystal characteristics imposed by the initial precursor solution as well as the processing parameters.

## 7. Conclusions and Future Recommendations

Nanocellulose can be extracted from cellulosic materials via the acid hydrolysis process in which the disordered regions are hydrolyzed by acid and the ordered portions are left. The self-assembly of CNC in aqueous solutions brings out many colors called iridescence due to its photonic structures. Nanocellulose is able to replace glass in optoelectronic devices by tailoring its optical properties. The pitch of CNC can be varied by introducing various electrolytes in it. CNC shows high transparency in the visible region. These optical properties are useful to make humidity sensors. The CNC has been found to be a green material to develop more functional materials that are birefringent. Additionally, some difficulties associated with nanocellulose have been identified. CNC necessitates high production costs, particularly at industrial levels. The chemical and energy consumption associated with the manufacture of CNC continues to be a bottleneck in the scale-up of nanocellulose production. However, to the best of our knowledge, progress has been made in this area, with many pilot-scale production facilities and a more environmentally friendly approach to CNC production now available worldwide. Nevertheless, the use of CNC in biomimetic applications is still limited. Therefore, more studies are needed to enhance the potential of CNC for other various fields. 

## Figures and Tables

**Figure 1 polymers-14-01546-f001:**
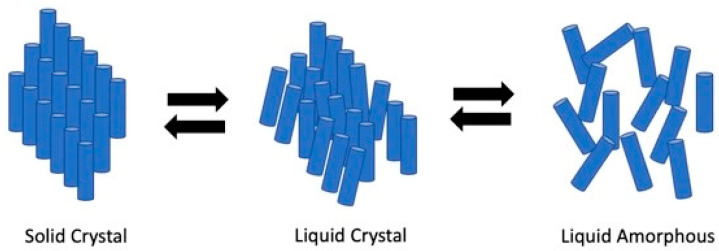
Illustration of rod-like particles of solid crystal, liquid crystals, and liquid phase.

**Figure 2 polymers-14-01546-f002:**
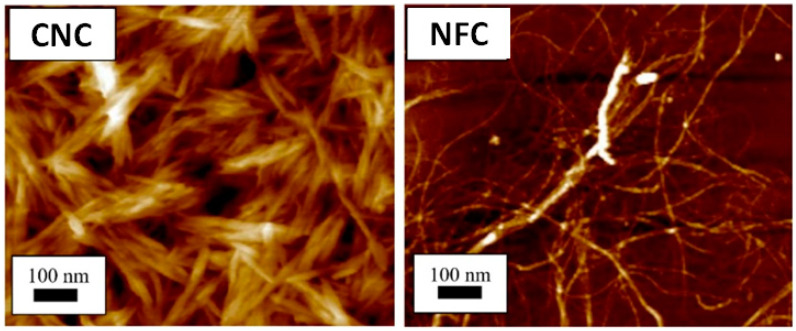
Atomic force microscopy images showing different structures of CNC [54] and NFC. Adapted from [55].

**Figure 3 polymers-14-01546-f003:**
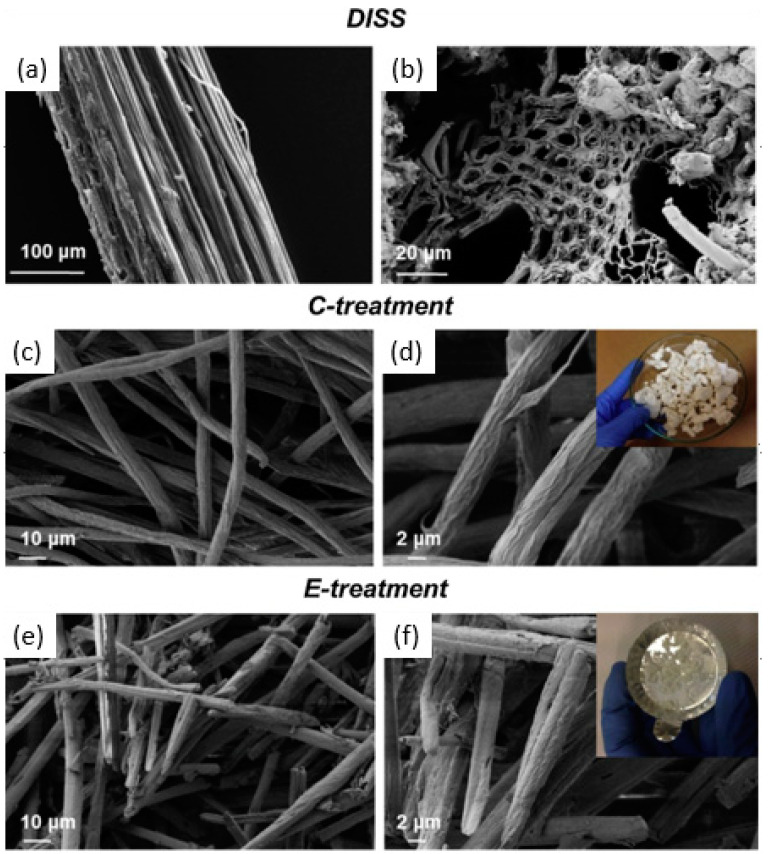
Morphological characterization of Diss fibers: longitudinal (**a**) and transversal section (**b**). FESEM investigation of bleached fibers using chemical (**c**,**d**) and enzymatic (**e**,**f**) treatment at two different magnifications. The visual image of treated fibers: insert (**d**) chemically treated fibers and insert (**f**) enzymatically treated fibers. Reproduced from [60].

**Figure 4 polymers-14-01546-f004:**
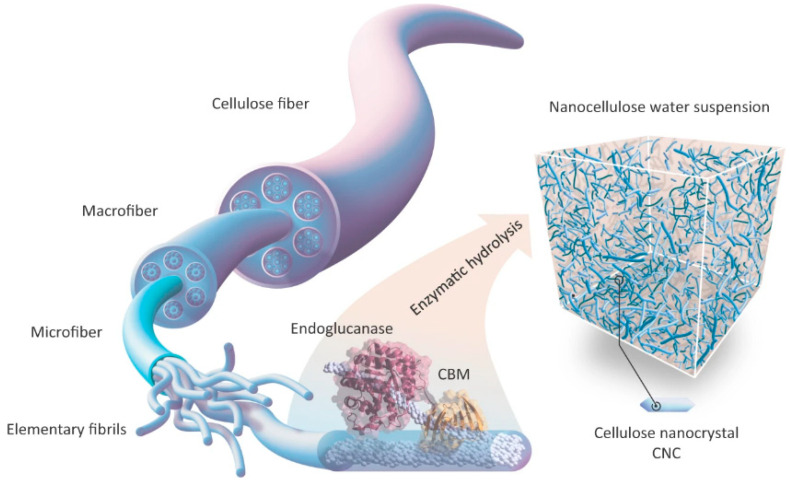
Illustration of formation of CNC via enzyme hydrolysis. Reproduced with permission from [73].

**Figure 5 polymers-14-01546-f005:**
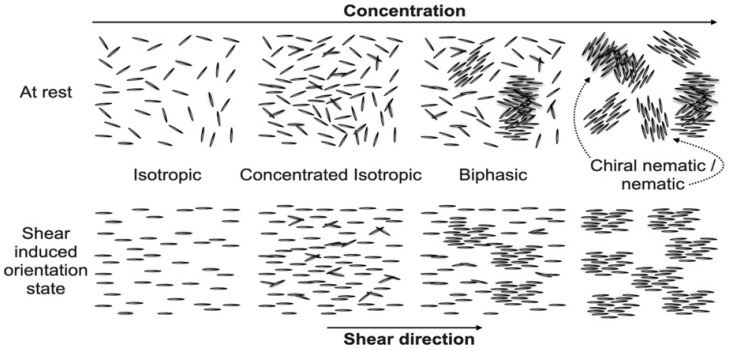
CNC self-assembly in a suspension at rest with rising concentration (**upper row**) and corresponding assembly in shear (**bottom row**).

**Figure 6 polymers-14-01546-f006:**
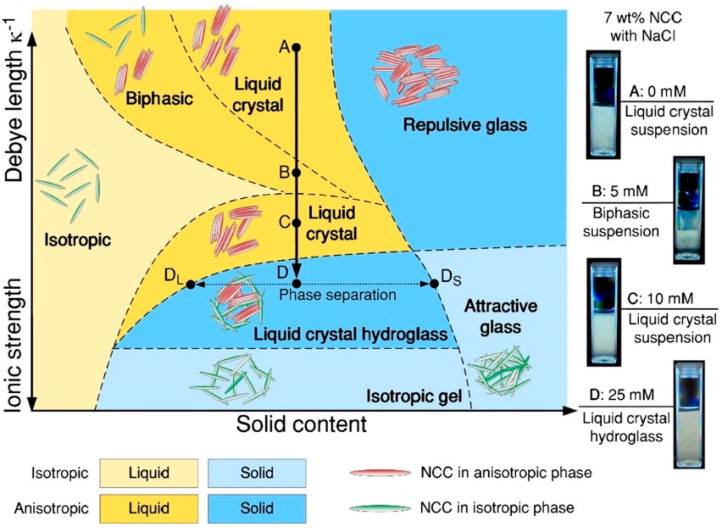
Schematic phase diagram of CNC aqueous suspension as a function of solid content and inter-particle attraction (represented by ionic strength). The relative positions of each phase and the geometry of the boundaries, as well as theoretical works in colloidal suspension gelation and liquid crystallinity, are from references [50,77,78,79]. Pictures A−D on the right were shot with cross-polarizers and corresponded to compositions A−D on the phase diagram, respectively. Reproduced from [75].

**Figure 7 polymers-14-01546-f007:**
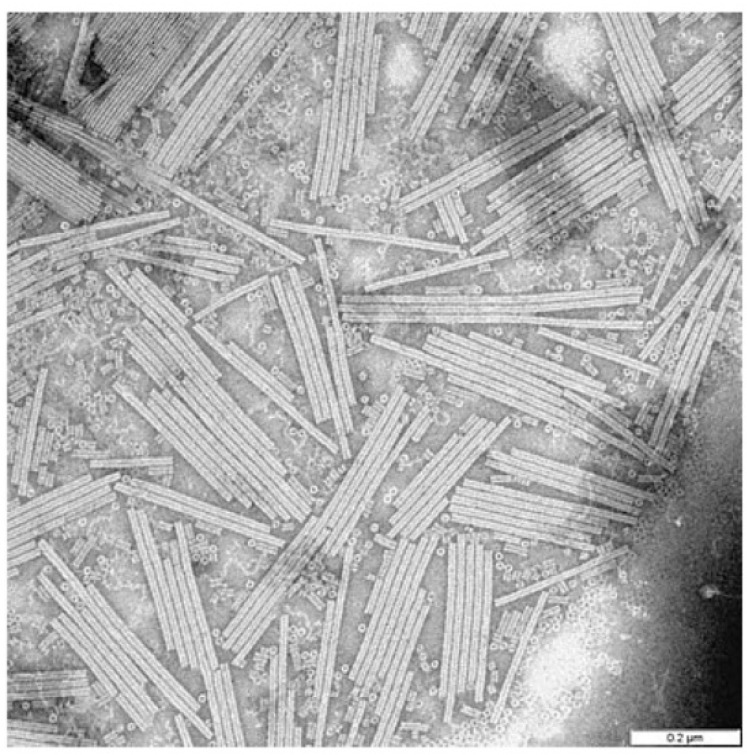
Cholesteric mesophase observed under electron microscopy. Reproduced from [80].

**Figure 8 polymers-14-01546-f008:**
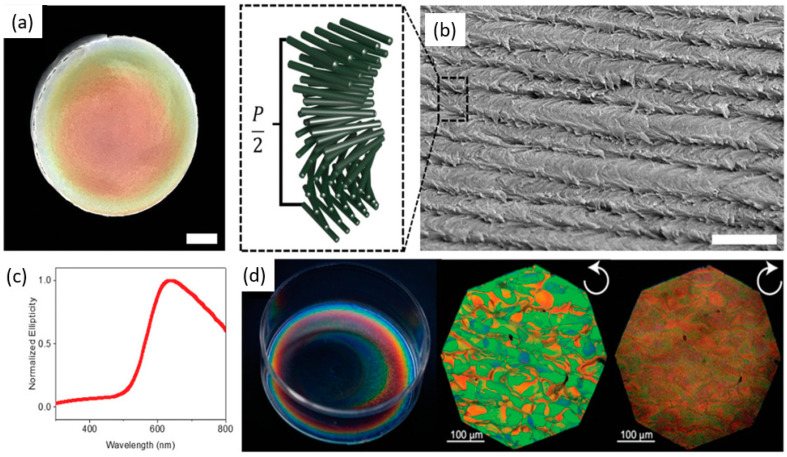
(**a**) A cholesteric CNC film’s structure and optical characteristics. (**b**) Photograph of chiral nematic CNC film displaying structural color iridescence. (**c**) Scanning electron micrograph of CNC in a film with a helical twist. The helical pitch is schematically represented in the dashed-line inset (P). (**d**) A typical cholesteric CNC film’s circular dichroism spectrum. Natural light (left), left-handed circularly polarized light (center), and right-handed circularly polarized light (right) views of chiral nematic CNC film (right). Reproduced from [81].

**Figure 9 polymers-14-01546-f009:**
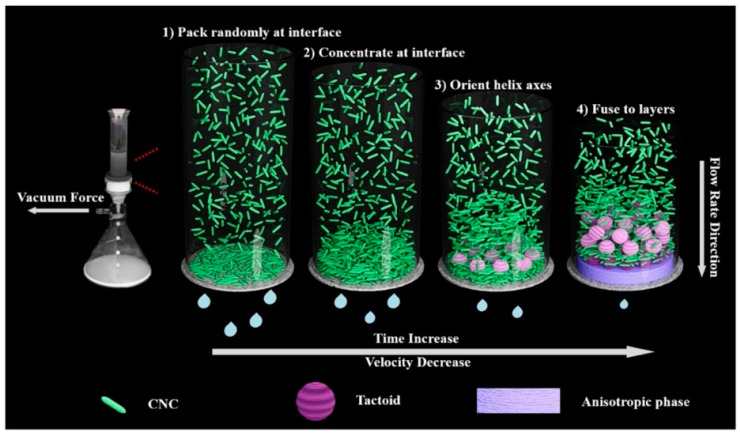
Schematic representation of the vacuum-assisted CNC self-assembly process. Reproduced from [82].

**Figure 10 polymers-14-01546-f010:**
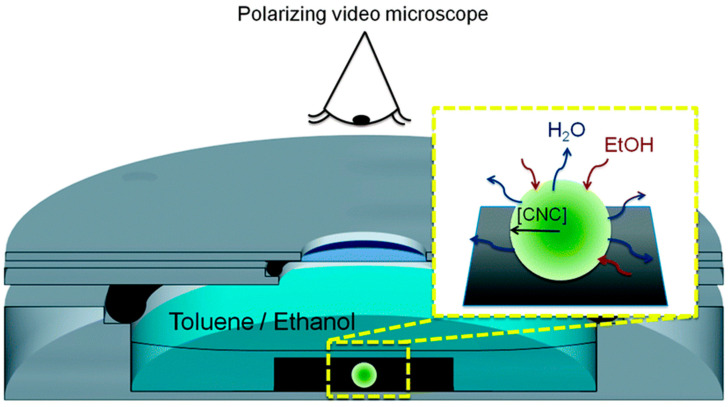
The cell utilized for polarized video microscopy imaging of the decreasing droplets is shown schematically. The CNC droplet (bright green) is captured in reflection mode on a hydrophobic OTS-silicon substrate (black). Water and ethanol diffusion and the resulting radial CNC concentration gradient are depicted. Reproduced from [89].

**Figure 11 polymers-14-01546-f011:**
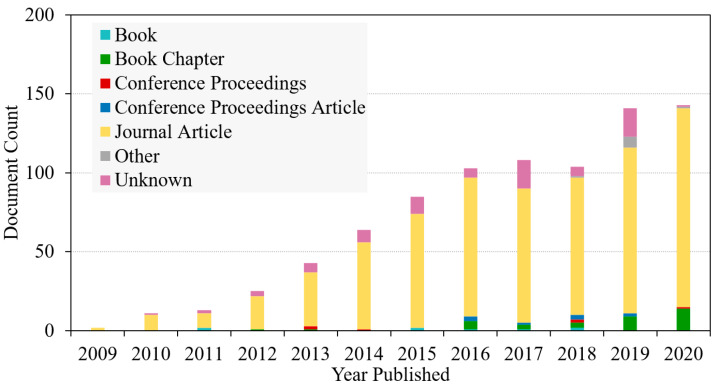
A chart of published manuscripts on the nanocellulose functionalization from Lens.org. accessed on 15 February 2021.

**Figure 12 polymers-14-01546-f012:**
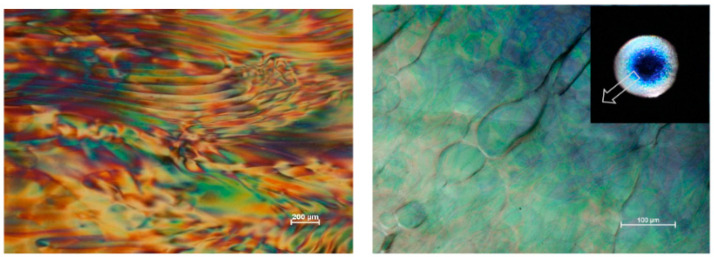
The 6% CNC suspension as viewed from polarized light microscopy [93].

**Figure 13 polymers-14-01546-f013:**
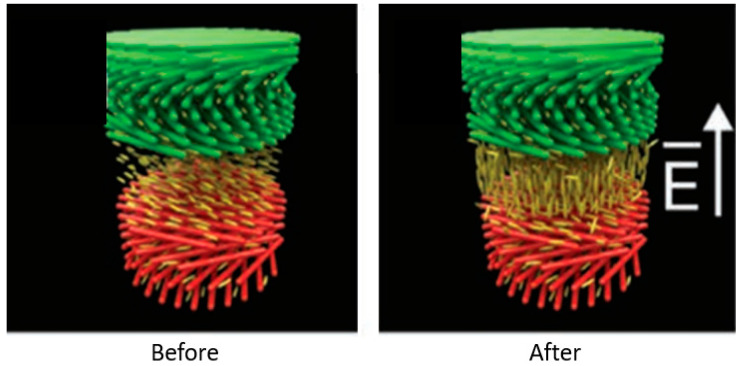
The orientational effect on CNC after an electric field (E) is applied to a 5CB positive dielectric nematic layer. Adapted from [108].

**Figure 14 polymers-14-01546-f014:**
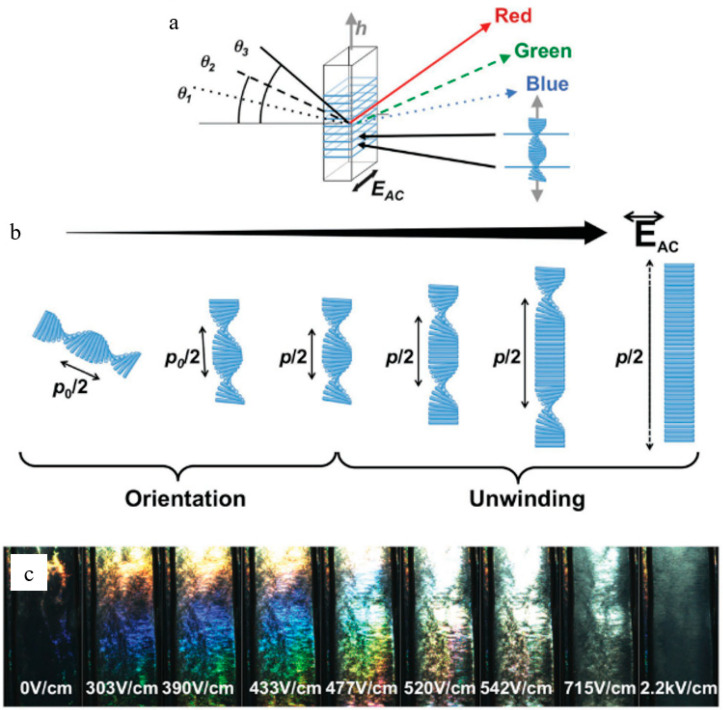
(**a**) Iridescence schemata, (**b**) cholesteric orientation and unwinding sequences upon the increase in the electric field, and (**c**) iridescence evolution upon electric field application (values in rms), demonstrating increased in light intensity, then redshift, and lastly color disappearance. Reproduced from [110].

**Figure 15 polymers-14-01546-f015:**
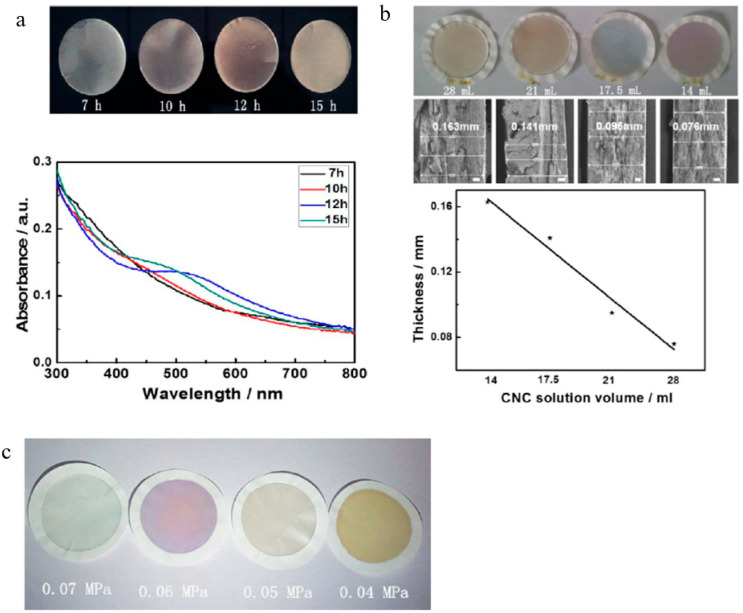
(**a**) Photographs of CNC iridescent solid films prepared with varying sonication times and their UV-Vis spectra; (**b**) optical photographs of CNC iridescent solid films prepared with varying CNC suspension volumes and their SEM images of the cross-sections, and a plot of film thickness as a function of CNC suspension volume; (**c**) photographs of CNC iridescent films prepared with various degrees of vacuum. Reproduced from [109].

**Figure 16 polymers-14-01546-f016:**
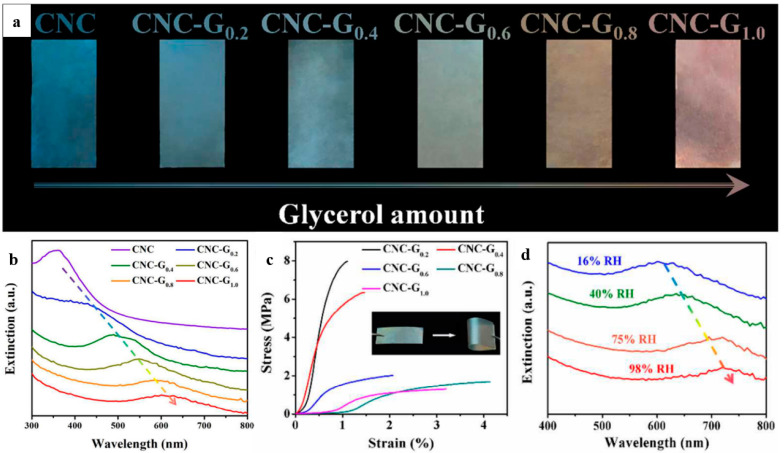
(**a**) Photographs of CNC films prepared with varying glycerol concentration; (**b**) UV-Vis spectra of the resultant CNC/glycerol films; (**c**) tensile property of CNC/glycerol films; and (**d**) effects of humidity-responsive CNC/glycerol films. Reproduced from [106].

**Figure 17 polymers-14-01546-f017:**
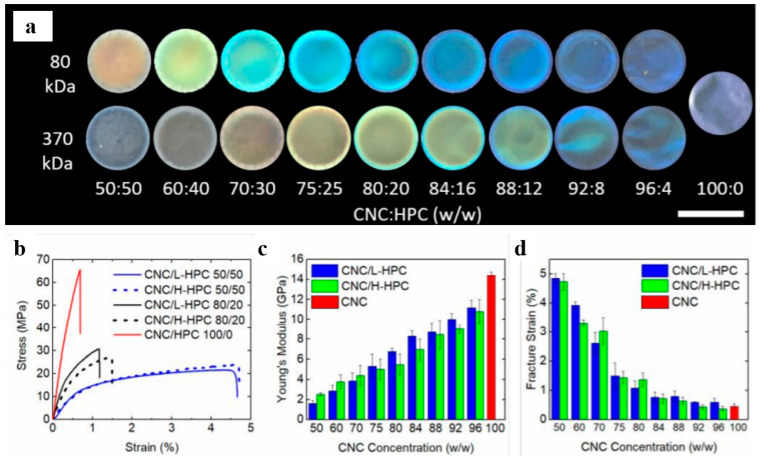
(**a**) Photographs of iridescent films prepared with varying molecular weight and HPC concentration (scale bar, 3.5 cm); (**b**) stress–strain curves for different CNC/HPC composite films; (**c**) Young’s modulus for different CNC concentrations for both H-HPC (370 kDa) and L-HPC (80 kDa) composite films; (**d**) fracture strains for different CNC concentrations for both H-HPC and L-HPC composite films. Reproduced from [112].

**Figure 18 polymers-14-01546-f018:**
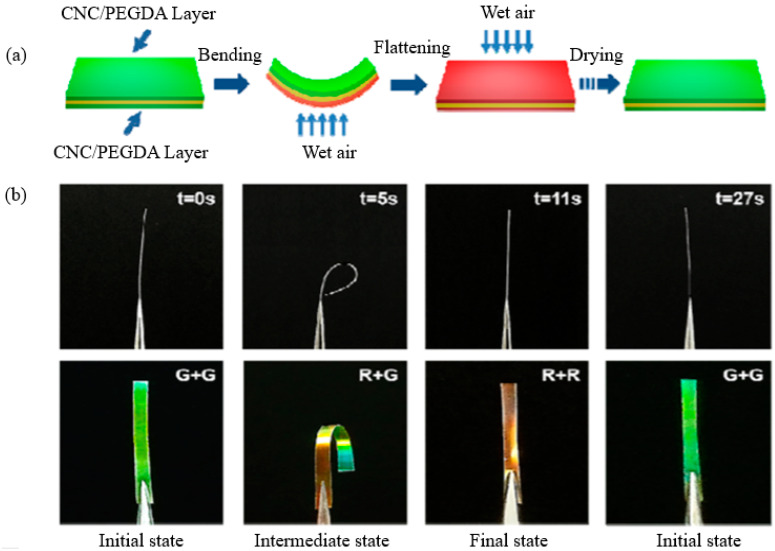
(**a**) Water vapor-based actuation in a sandwiched structure nanocomposite film is shown schematically; (**b**) images of the sandwiched nanocomposite film deformation and coloration. Reproduced from [116].

**Figure 19 polymers-14-01546-f019:**
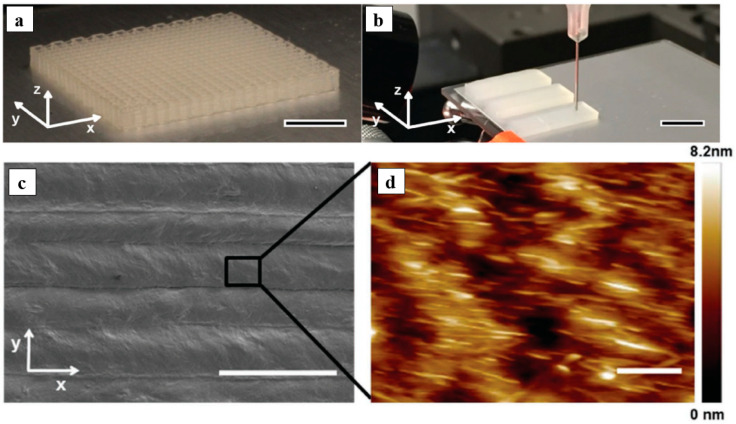
(**a**,**b**) Photograph of 3D printed grids and blocks with eight layers of parallel lines; (**c**,**d**) CNC alignment in 3D printed films: SEM (scale bar, 200 µm) and AFM topologies (scale bar, 100 µm). Reproduced from [118].

**Figure 20 polymers-14-01546-f020:**
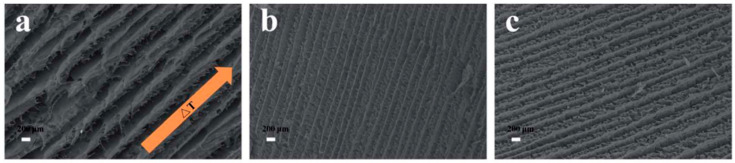
(**a**–**c**) Using the directionally frozen approach, SEM images of the longitudinal side views of freeze-casted porous foams were identified on the long axis. Reproduced from [121].

**Figure 21 polymers-14-01546-f021:**
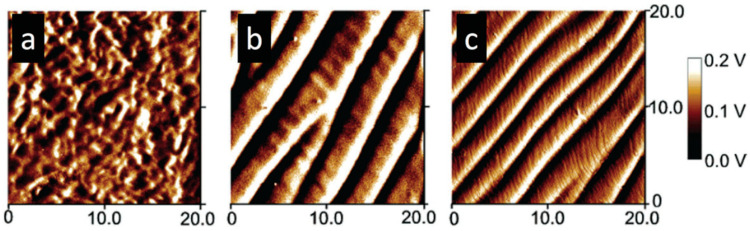
AFM top-view images CNC/HPC sheared films (shear rate 5 mm s^−1^) from (**a**) isotropic and (**b**,**c**) liquid-crystalline solutions. Reproduced from [122].

**Table 1 polymers-14-01546-t001:** List of CNC from various cellulose sources by sulfuric acid hydrolysis.

Cellulose Sources	Length (nm)	Width (nm)	Ref.
Bacteria	100–1000	10–50	[63]
Cotton	70–300	5–10	[64]
Wood	100–300	3–5	[65]
Tunicate	500–1000	10–30	[66]
Valonia	>1000	10–20	[67]

**Table 2 polymers-14-01546-t002:** List of previous reports regarding enzymatic hydrolysis in CNC production.

Cellulose Sources	Before Enzyme Hydrolysis	After Enzyme Hydrolysis	Enzyme	Dimensions and Crystallinity (CI)	Ref.
Sugarcane straw	Chemical treatment	-	Cellic^®^ CTec3	D: 8.7–14.1 nm L: 395.5–507.7 nm CI: 66.7–70.4%	[68]
Eucalyptus cellulose kraft pulp	Ball milling	Sonication	On-site production by *A. niger* strain	D: 24 nm L: 294 nm CI: 77.9–78.3%	[69]
Bleached Eucalyptus kraft pulp		Sonication	Monocomponent EGs	D: 6–10 nm L: 400–600 nm CI: 88%	[70]
Sugarcane bagasse	Steam explosion/liquid hot water	Chemical treatment; acid hydrolysis	Cellic^®^ CTec2	D: 14–18 nm L: 195–250 nm CI: 77.7%	[71]
Wheat microcrystalline cellulose	Sonication	-	Celluclast 1.5 L	D: <10 nm L: 40–200 nm CI: 74.4–87.5%	[72]

## Data Availability

The data presented in this study are available on request from the corresponding author.

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
