# Peer review of "Emerging Developments on Nanocellulose as Liquid Crystals: A Biomimetic Approach"

_polymers, 2022, doi:10.3390/polym14081546_

Round 1
Reviewer 1 Report
It is an excellent review article on bioinspired materials. Comprehensive information covered all aspect including bimimic, LC, extraction, fabrications, and characterisations.
Author Response
Thank you for your comment.
Reviewer 2 Report
The authors present a review of recent developments of the applications of nanocellulose liquid crystals. Specifically, the biomimetic aspect and optical device applications of these materials. They discuss the extraction and isolation of cellulose nanocrystals (CNCs) and well as a brief introduction of the class of materials. Then they discuss the development of liquid crystals and their applications: thin films and biomimetics. The review is well organized and will be of interest to abroad audience. The work provides sufficient background that scientists of many diverse backgrounds can understand and appreciate the work. It is recommended for publication after minor review. The minor issues are given as following:
- Minor English use issues
Line 77: “generating a lot of buzz”
Line 116: “”has gained” “scientists”
Line 287: “coalesced”
Line 634: “has also been discovered” same with Line 645
- Line 100. Is acid hydrolysis only done with sulfuric acid? The next sentence mentions sulfur group with negative charge. Can other acids be used?
- Line 138: I think this paragraph should be expanded to include all topics the review is covering such as the biomimetic application
- Line 271: Remove (2020) from the in text citation
- Line 340: This sentence requires some reference
- Line 387: “UF” has not been defined
- Line 417: This sentence requires some reference
- Line 649: polyethylene glycol has already been defined as PEG
Author Response
Dear respected reviewer,
We had corrected the manuscript based on all your comments. All changes were highlighted in yellow. Responses to the comments are listed below.
Thank you.
Comment 1
Minor English use issues
Line 77: “generating a lot of buzz”
Line 116: “has gained” “scientists”
Line 287: “coalesced”
Line 634: “has also been discovered” same with Line 645
Response: Thank you. We had corrected all the listed issues.
Comment 2
Line 100. Is acid hydrolysis only done with sulfuric acid? The next sentence mentions sulphur group with negative charge. Can other acids be used?
Response: According to Ahmad and Pant, (2018), hydrochloric acid, sulfuric acid, trifluoroacetic acid, formic acid, and nitric acid are commonly used in acid hydrolysis. However, sulfuric acid hydrolysis has been recognized as the method most widely used for preparing CNCs because the process is simple and results in nanoparticle (100–1000 nm) with highly crystalline and stiff (Kusmono et al., 2020). We had included this information in the revised manuscript.
Comment 3
Line 138: I think this paragraph should be expanded to include all topics the review is covering such as the biomimetic application
Response: Thank you. We had revised the paragraph.
Comment 4
Line 271: Remove (2020) from the in-text citation
Response: Thank you. We had removed it.
Comment 5
Line 340: This sentence requires some reference
Response: Thank you. We had included the reference.
Comment 6
Line 387: “UF” has not been defined
Response: Thank you. We had revised the term.
Comment 7
Line 417: This sentence requires some reference
Response: Thank you. We had included the reference.
Comment 8
Line 649: polyethylene glycol has already been defined as PEG
Response: Thank you for your comment. We had revised the term.